# Hepatitis B Virus Seroprevalence in Ayacucho, Peru: A Comprehensive Review Across the Pre-Vaccination and Post-Vaccination Periods

**DOI:** 10.3390/vaccines13090916

**Published:** 2025-08-28

**Authors:** Homero Ango-Aguilar, Jimmy Ango-Bedriñana, Obert Marín-Sánchez, Ruy D. Chacón

**Affiliations:** 1Programa Académico de Microbiología, Facultad de Ciencias Biológicas, Universidad Nacional de San Cristóbal de Huamanga, Portal Independencia N° 57-Huamanga, Ayacucho 05001, Peru; homero.ango@unsch.edu.pe; 2Escuela Profesional de Medicina Humana, Universidad Nacional de San Cristóbal de Huamanga, Portal Independencia N° 57-Huamanga, Ayacucho 05001, Peru; jimmy.ango@unsch.edu.pe; 3Instituto de Investigación de Ciencias Biológicas Antonio Raimondi (ICBAR), Facultad de Ciencias Biológicas, Universidad Nacional Mayor de San Marcos, Av. Carlos Germán Amezaga 375, Lima 15081, Peru; omarins@unmsm.edu.pe; 4Departamento Académico de Microbiología Médica, Facultad de Medicina, Universidad Nacional Mayor de San Marcos, Av. Carlos Germán Amezaga 375, Lima 15081, Peru; 5Department of Pathology, School of Veterinary Medicine, University of São Paulo, Av. Prof. Orlando M. Paiva, 87, São Paulo 05508-270, Brazil; 6Pathogen Genetics Research Group (PATHO-GEN), OMICS, Lima 15001, Peru

**Keywords:** viral hepatitis, hepatitis B virus, HBsAg, anti-HBc, vaccination, Ayacucho, Peru, blood donors, pregnant women

## Abstract

Hepatitis B virus (HBV) infection remains a global public health concern, with perinatal transmission as the primary route in endemic populations. Ayacucho is a priority region due to its high incidence (second nationally between 2019 and 2024) and the significant decline in vaccination coverage (~15%). This study aims to synthesize existing epidemiological evidence on HBV seroprevalence in Ayacucho, Peru, emphasizing temporal changes observed before and after the implementation of vaccination programs to inform control strategies. This review was conducted, integrating data from diverse population groups, including children, pregnant women, blood donors, high-risk individuals (military personnel, female sex workers, prisoners), and household contacts, to identify transmission patterns and evaluate the impact of immunization efforts. Historically, Ayacucho was hyperendemic, with an HBsAg prevalence of 20% in Huanta (1985–1986) and a high mortality from liver diseases. The introduction of a vaccination in the 1990s led to a drastic reduction in infection rates among children, from 24.4–30.4% (1994) to 2.3–5.1% (1997), and improved overall Expanded Program on Immunization (EPI) coverage. However, recent data (2000–2024) reveal a concerning increase in HBV cases since 2012, with peaks in 2016 and 2023, correlating with a decline in vaccination rates post-2021. HBV prevalence remains elevated among high-risk populations—including military personnel, female sex workers, and prisoners—as well as among blood donors (HBsAg: 3.73–5.0%; anti-HBc: 21–33%). In addition, significant knowledge gaps and low adherence to EPI strategies were observed. Despite initial vaccination success, Ayacucho faces a resurgence of HBV infection, exacerbated by declining vaccine coverage and vulnerabilities in high-risk populations. Reinforcing immunization programs and screening strategies is urgent to control and eventually eliminate HBV cases in the region.

## 1. Introduction

Viral hepatitis is a communicable disease that causes both acute and chronic illness and significantly contributes to rising global mortality. In 2022, approximately 1.3 million people died from viral hepatitis [1]. Five main hepatitis viruses—A, B, C, D, and E—have been identified as the primary causes of infection and disease, each with distinct epidemiological characteristics across different regions and populations worldwide [1]. Among them, hepatitis B virus (HBV) accounts for the greatest burden in terms of disease severity and mortality. It is estimated that 254 million people are currently living with chronic HBV infection, and over 1 million deaths occur annually due to complications such as chronic liver disease, cirrhosis, and hepatocellular carcinoma. However, only about 13% of individuals with chronic HBV have been diagnosed, and fewer than 3% have received antiviral therapy [1].

HBV is a small, enveloped DNA virus of the family *Hepadnaviridae*, genus *Orthohepadnavirus*, and is classified under the species *Orthohepadnavirus hominoidei* [2]. Notably, HBV exhibits substantial genetic diversity and is categorized into at least ten genotypes (A–J), each with distinct geographic distributions and variable associations with clinical outcomes and treatment responses [3].

Serological diagnosis of HBV relies on the detection of specific viral antigens and host antibodies using immunological methods such as enzyme-linked immunosorbent assays (ELISA), chemiluminescence immunoassays (CLIA), and rapid diagnostic tests. Key serological markers include the hepatitis B surface antigen (HBsAg), antibody to surface antigen (anti-HBs), total and IgM antibodies to the core antigen (anti-HBc total/IgM), hepatitis B e antigen (HBeAg), and its corresponding antibody (anti-HBe) [4,5]. Interpretation of these markers enables the classification of individuals into different stages of HBV infection. In acute infection, both HBsAg and anti-HBc IgM are positive, and HBeAg may also be detectable. In chronic infection, HBsAg remains positive for more than six months, anti-HBc (typically total) is positive, and HBeAg may be present or absent depending on the replication phase. A past (resolved) infection is indicated by the presence of anti-HBc (total) and anti-HBs, with a negative HBsAg. Individuals who have been successfully vaccinated against HBV will show only anti-HBs positivity, with all other markers negative [4,5].

The persistence and pathogenesis of hepatitis B virus (HBV) are influenced by multiple viral and host factors, including genotype distribution, viral mutations, and the host immune response [3]. Approximately two-thirds of people infected with HBV live in high-endemicity areas, including sub-Saharan Africa, Asia, and the Pacific, with prevalence rates reaching nearly 20% in countries such as Chad (19.3%) and South Sudan (18.1%) [6,7]. In these regions, individuals are often infected at early ages, and up to 90% progress to chronic infection and hepatocellular carcinoma [6].

The introduction of HBV vaccination programs has led to a significant global decline in infection rates, particularly among pediatric populations. According to the World Health Organization, the prevalence of hepatitis B surface antigen (HBsAg) among children under five years of age has fallen below 1% in areas with high vaccine coverage [1]. However, the success of these programs remains uneven across countries and within regions of the same country, often reflecting disparities in public health infrastructure, socioeconomic conditions, and program implementation effectiveness [8].

Historically, until the 1990s, Peru was classified as a country with intermediate HBV endemicity, with reported seroprevalence rates ranging from 2% to 7%, based on averages from various regions [9]. Following the implementation of widespread vaccination programs by the Peruvian Ministry of Health across both urban and rural areas, Peru transitioned to a low-endemicity country [10,11]. Nonetheless, prevalence rates have varied significantly across the country’s three main geographical regions, and even within those regions at the local level. For example, in the Amazon jungle, endemicity ranged from moderate to high, with seroprevalence rates of 2.5% in the general population of Iquitos, reaching up to 20% in certain Indigenous communities [10,12]. Along the coast, reported rates ranged from 1% to 3.5% [10,13]. In the Andean highlands, studies have shown low to high endemicity, including in Ayacucho, where prevalence rates reached as high as 9.8% three decades ago [10,14].

Therefore, this review aims to comprehensively synthesize the existing epidemiological evidence on hepatitis B virus (HBV) seroprevalence in Ayacucho, Peru, with an emphasis on the temporal changes observed before and after the implementation of HBV vaccination programs. By chronologically integrating data from diverse population groups—including children, pregnant women, blood donors, high-risk individuals, and household contacts—this review seeks to identify patterns of transmission, measure the long-term impact of immunization efforts, and highlight persistent vulnerabilities in prevention and control strategies. The findings presented herein are intended to inform public health decision-making and guide the design of region-specific interventions toward HBV elimination.

## 2. Epidemiological Overview of Hepatitis B Virus in Peru

Hepatitis B virus (HBV) remains a significant public health concern in Peru, marked by notable regional disparities in incidence. To analyze the epidemiological overview and dynamics of HBV confirmed cases in Peru, we used publicly available information from official sources of the Peruvian Ministry of Health, accessible through the electronic portal for public health surveillance [15]. This was supplemented with documents provided by the General Directorate of Strategic Public Health Interventions (DGIESP), including disaggregated epidemiological data on hepatitis B and vaccination campaign records (Appendix A). The data selected and analyzed correspond exclusively to confirmed cases. Confirmed cases are defined as those with a positive serological result compatible with active infection (at least positive for the HBsAg marker) and include both acute and chronic infections. Additionally, population data were obtained from the electronic portal of the National Institute of Statistics and Informatics (INEI) of Peru [16].

Between 2019 and 2024, the average annual HBV case rates per 100,000 inhabitants varied considerably across the country (Figure 1). The highest rates were observed in the Amazon and Andean regions, with Ucayali, Ayacucho, and Cusco leading the list. These findings underscore the presence of transmission hotspots and emphasize the need for targeted epidemiological surveillance and region-specific public health interventions.

A closer examination reveals that the regions with the highest average HBV incidence were Ucayali and Ayacucho, followed by Cusco, Pasco, Huancavelica, and Madre de Dios (Figure 2). The top ten regions are predominantly located in the Amazonian and Andean zones, areas historically associated with HBV endemicity and known for logistical challenges in healthcare access. This ranking underscores the pronounced regional disparities in HBV burden and identifies priority areas for targeted prevention and control interventions.

Vaccination coverage is a critical factor influencing HBV transmission dynamics in Peru. Between 2020 and 2024, several regions experienced substantial declines in HBV vaccine target coverage (Figure 3). Notably, regions such as Huancavelica, Apurímac, Ayacucho, and Pasco reported reductions exceeding 10%. Some of these same regions also exhibited the highest HBV incidence rates, suggesting a potential association between declining immunization levels and increased HBV transmission.

In this review, Ayacucho was selected as the focal region for further investigation due to its epidemiological burden and programmatic vulnerability. It ranked second nationwide in HBV incidence between 2019 and 2024 and experienced one of the most significant declines in vaccine target coverage (approximately −15%) during the same period. This combination of high disease burden and weakened prevention efforts positions Ayacucho as a priority setting for HBV control strategies and research.

## 3. Historical Evidence and Seroprevalence in the General Population of Ayacucho

One of the earliest comprehensive assessments was conducted through a retrospective analysis of death certificates spanning 32 years (1960–1982) in the municipality of Huanta, the northernmost province of Ayacucho and historically one of the most affected by hepatitis. Among 7996 recorded deaths, 8.2% were attributed to liver-related conditions. Fulminant hepatitis was the leading cause (55%), followed by liver cirrhosis (36%) and hepatocellular carcinoma (9%), indicating a substantial burden of progressive liver disease during that period [17,18].

In 1979, a review of 915 clinical records from Huanta Hospital (covering 1967–1976) identified hepatitis as an endemo-epidemic condition in the region. Males under 20 years of age were disproportionately affected, with an overall case-fatality rate of 14.75%, notably higher in males (10.3%) than in females (4.45%). Monthly trends revealed a consistent pattern: 80% of clinically diagnosed patients recovered, while 20% succumbed to the disease [19].

Further evidence emerged in 1988 from a systematic review of 6179 medical records and statistical reports from Huanta Support Hospital, covering the period 1982–1986. Among these, 430 cases (6.69%) were diagnosed as viral hepatitis, with infectious hepatitis (presumably hepatitis A) accounting for 83.93% of cases and serum hepatitis (likely HBV or HCV) representing 16.07%. Most affected individuals were juveniles (170 cases) or children under five years of age (118 cases). Males accounted for 64.3% of cases, compared to 35.5% for females. Cirrhosis was reported as a complication in 7.9% of cases, and the overall mortality rate for viral hepatitis was 5.23% [20]. Together, these studies not only raised national awareness of the hepatitis burden in Ayacucho but also laid the scientific foundation for subsequent investigations using serological and molecular markers, thereby advancing the epidemiological understanding of viral hepatitis in the region.

The first clear seroepidemiological evidence of hepatitis B virus (HBV) prevalence in Huanta was obtained between 1985 and 1986 through the analysis of blood samples from patients clinically diagnosed with hepatitis. The study reported a notably high HBsAg prevalence of 20%, with the highest rates observed among individuals aged 11–20 years [21]. These findings suggested probable iatrogenic transmission linked to the reuse of non-sterile syringes and instruments, highlighting the role of unsafe medical practices in the dissemination of HBV during that period.

This evidence supported the decision of the Peruvian Ministry of Health to initiate a pilot HBV vaccination program between 1991 and 1994 in the inter-Andean valleys of the highlands, including Huanta and Abancay. As a result, HBV infection rates in children declined significantly—from 24.4–30.4% in 1994 to 2.3–5.1% in 1997 [22]. The program was subsequently expanded in 1996 to include children under one year of age living in intermediate- and high-endemic areas, and by 2003, universal HBV vaccination was implemented nationwide for all children under one year of age. The long-term impact of these interventions remained evident in the years following the 2000s [10].

From 2014 to 2015, a nationwide study confirmed Peru’s classification as a low-HBV-endemicity country. However, within this study, Ayacucho stood out with a 1.7% HBsAg prevalence and the nation’s highest anti-HBc positivity (51.7%), suggesting ongoing viral circulation in the region [10].

A subsequent cross-sectional seroprevalence study in 2017, conducted in Luricocha, Ayacucho, included 224 residents aged 5 to 75 who had lived there for at least three years. This study found five individuals positive for HBsAg, resulting in a 2.2% prevalence. Males showed a slightly higher rate (2.4%) compared to females (2.1%) [23]. These figures position the region at the lower boundary of intermediate HBV endemicity.

Furthermore, historical data from 2000 to 2024, gathered from the National Center for Epidemiology, Prevention and Disease Control—MINSA (Ministry of Health of Peru), consolidates 25 years of information and reveals a concerning upward trend in HBV incidence in the region (Figure 4). Although some year-to-year fluctuations were observed, there was a sustained increase in the HBV incidence after 2012, with notable peaks in 2016 (427 annual cases) and 2023 (224 annual cases). It is important to note that the expansion of HBsAg screening and the adoption of more sensitive HBsAg tests may have contributed to the increased number of registered cases. In late 2018, Peru introduced HBV testing for pregnant women and individuals at high risk of HBV infection through a Ministry of Health Resolution (NTS No. 146-MINSA/2018/DGIESP) [24]. This policy was aligned with World Health Organization (WHO) guidelines on the prevention, care, and treatment of persons living with chronic hepatitis B (CHB) [25,26]. Alongside the expansion of screening in various at-risk populations, there has also been a gradual implementation of more sensitive detection methods such as ELISA and chemiluminescence immunoassays [10,27]. In any case, these findings indicate ongoing viral transmission despite previous interventions and underscore the urgent need to strengthen surveillance and control measures.

## 4. Seroprevalence in Children

In 1994, a cross-sectional study in Huanta investigated HBV serological markers in 333 clinically healthy children under six years of age (183 boys, 150 girls). The results revealed an 8% prevalence of HBsAg (indicating active infection) and 29% prevalence of anti-HBc (suggesting past infection). HBV infection prevalence (anti-HBc) progressively increased with age: 10.7% at one year, 16.6% at two years, 18.9% at three years, 20.8% at four years, 32.0% at five years, and 48.5% at six years. No significant gender difference was observed. Notably, 16.4% of HBV-infected children also tested positive for anti-HDV, suggesting early-life co-infection. The age-related increase in HBV infection, coupled with the lack of significant associations with traditional risk factors, supports the hypothesis of horizontal transmission within this population [28,29,30].

In the same year (1994), another study in Huanta analyzed HBV markers in 143 clinically healthy schoolchildren (7–20 years old) randomly selected from four public schools. Findings indicated high endemicity, with 16% testing positive for HBsAg and 82% for anti-HBc, further highlighting widespread HBV exposure in the region during that period [31].

Between 1999 and early 2000, an expanded study examined 716 children aged 1 to 14 years from the rural areas of the Luricocha district (Huanta), comprising 341 girls and 375 boys. The study identified 4.9% (n = 35) as chronic carriers of HBsAg, with 20 boys and 15 girls among them. Among the 34 children evaluated in greater detail, 100% were anti-HBc positive, and 29.4% tested positive for HBeAg, indicating active viral replication. HBsAg prevalence increased from five years of age onward, while anti-HBc positivity increased from three years of age, demonstrating ongoing community-level transmission in this rural setting [32].

Finally, in 2002, a seroepidemiological survey (n = 130) was conducted among school-aged children (5–11 years old) from five rural highland communities on the left bank of the Pampas River (Llaweccmarca, Vacahuasi, Ninabamba, Ayrabamba, and Hatuncusi), located between 2300 and 2600 m above sea level. The study reported that 11.4% were HBsAg carriers, and 47.7% had evidence of past infection (anti-HBc positivity). No significant differences were found by sex or age group [33].

Recent post-COVID-19 data indicates a concerning upward trend in the incidence of HBV among children, with Ayacucho reaching a peak of 9.19 cases per 100,000 inhabitants (Figure 5). This increase is mirrored in adolescents, whose incidence rates peak between 2.56 and 6.17 per 100,000 inhabitants, suggesting that the rise in HBV cases is not confined to a single age group. The observed trends may reflect the broader consequences of disruptions in routine vaccination programs during the pandemic, which likely reduced herd immunity and increased susceptibility among younger populations. These patterns underscore the urgent need to reinforce vaccination coverage and public health interventions, as even modest declines in immunization can facilitate the reemergence of preventable infections. Furthermore, the data highlights the importance of continuous surveillance to detect early shifts in HBV epidemiology and guide targeted strategies to mitigate transmission, particularly in regions with historically medium to high endemicity.

## 5. Seroprevalence in Pregnant Women

A study conducted in 1996 and 1997 included 126 pregnant women, with a mean age of 24.9 years (range: 15 to 48), who attended the General Hospital of Huanta for routine prenatal care. The prevalence of HBsAg was 3.2%, while 73.0% tested positive for anti-HBs. This study also assessed pregnant women in Apurimac (1.36% for HBsAg and 34.8% for anti-HBs), Junin (1.38% for HBsAg and 16.4% for anti-HBs), and Lima (0.38% for HBsAg and 2.6% for anti-HBs). This research not only demonstrated the highest prevalence of HBV in pregnant women from Ayacucho, but also suggested higher exposure to HBV infection in the region. However, the study did not differentiate between natural infection and vaccination response [34].

Another study, conducted between August and October 1999, analyzed 181 serum samples from pregnant women aged 16 to 46 years attending the Gynecology and Obstetrics outpatient clinics of the Regional Hospital of Ayacucho. The seroprevalence of the hepatitis B surface antigen (HBsAg) was 1.65% (n = 3). Identified epidemiological risk factors included early onset of sexual activity, multiple sexual partners, unprotected sexual intercourse, and overcrowded living conditions. Additionally, the study reported a seroprevalence of 0.55% for HIV and 0.0% for *Treponema pallidum* [35].

More recently, a study conducted at the San Francisco Hospital in the VRAEM region of Ayacucho between 2018 and 2021 focused on pregnant women diagnosed with chronic hepatitis B. The incidence of chronic hepatitis B cases increased annually, from 0.5% in 2018 to 2.1% in 2021. Affected women were primarily aged between 19 and 43 years, with 44.8% belonging to the 26–30 year age group. Most participants were from rural areas (64.7%), identified as housewives (75%), cohabiting with their partners (75.86%), and reported initiating sexual activity between ages 12 and 17 (73.3%). The majority reported having had one or two sexual partners and engaging in unprotected sex (88.8%). Additionally, 8% had a family history of hepatitis B-related death, 7% had liver cirrhosis, and 6% reported a maternal history of hepatitis B. Alarmingly, 98% of the pregnant women in this study were unvaccinated [36].

Data from the last seven years from the National Center for Epidemiology, Prevention and Disease Control—MINSA (Ministry of Health of Peru) revealed an oscillation in the HBV incidence number in pregnant women, with a notable peak in 2022 (71 annual cases) (Figure 6). These findings are consistent with the general Ayacucho population (Figure 4), confirming an increasing trend in the number of cases in recent years.

## 6. Seroprevalence in Blood Donors

Blood transfusion remains a potential route for hepatitis B virus (HBV) transmission, especially in hyperendemic regions. In Peru, mandatory serological screening for HBsAg and anti-HBc has significantly reduced the risk of transfusion-transmitted HBV. However, in highly endemic regions like Ayacucho, the availability of eligible donors is severely limited, as 80–90% of potential blood donors may test positive for anti-HBc, indicating past infection and possible occult HBV.

In 2001, a serological screening of 169 blood donors was conducted at the Regional Hospital of Ayacucho. Testing included screening for HIV-1/2, HBV (HBsAg and anti-HBc), HCV, HTLV-I/II, *Trypanosoma cruzi*, and *Treponema pallidum*. Results showed that 3.73% were HBsAg positive and 33% were anti-HBc positive, suggesting previous exposure to HBV. Positivity was more frequent in males aged 20–30 years, mostly from Huamanga and Huanta. Among donors from Huamanga (n = 104), the prevalence was 1.16% for HBsAg and 21.49% for anti-HBc. Additional pathogen prevalence included 1.40% for HCV, 0.70% for HTLV, and 0.23% for HIV and *T. pallidum* [37].

This study continued in 2022 with 269 additional donors, revealing a 4.09% prevalence for HBsAg and 23.42% for anti-HBc. Age-based analysis showed higher HBV marker prevalence among donors aged 18–26 years, with 3.35% HBsAg and 9.29% anti-HBc positivity. Again, male donors showed higher prevalence than females. Most donors resided in the Huamanga province, where prevalence rates were 2.23% for HBsAg and 8.92% for anti-HBc. Anti-HCV prevalence in this cohort was 1.12% [38].

In a 2009 study, the focus shifted to evaluating eligibility criteria and the frequency of reactive markers among 727 blood donation candidates at the same hospital. Of these, 656 individuals (90.2%) were deemed eligible, while 71 (9.8%) were excluded. Among eligible donors, 22.6% (n = 147) had reactive serological markers, and 77.6% (n = 509) were non-reactive. Notably, among apparently healthy individuals of both sexes, 63.6% tested positive for HBsAg, a surprisingly high rate that reinforces the need for strict donor selection protocols in endemic settings [39].

In 2018, a further study was conducted at the Miguel Ángel Mariscal Llerena Regional Hospital of Ayacucho to assess the prevalence of HBV markers among 400 prospective blood donors of both sexes. The results showed that 5% tested positive for HBsAg and 21% for anti-HBc, while 74% tested negative for both markers. The seroprevalence of HBV was higher in males (17%), and most seropositive individuals were aged 28–37 years, among whom 10% tested positive. Additionally, 17% of seropositive candidates were single and 20% lived in urban areas, the majority of whom were students [40].

## 7. Seroprevalence in Household Contacts

While sexual, parenteral, and perinatal routes of hepatitis B virus (HBV) transmission are well established, these classical pathways do not seem to fully explain most infections in certain populations [41]. This is particularly true for children in hyperendemic areas and in low-endemic settings that receive migrants from high-endemicity zones. Instead, horizontal and intrafamilial transmission, especially in households with chronic carriers and susceptible individuals, have been proposed as more relevant routes [42].

Perinatal transmission typically results in chronic infection in about 90% of cases [6]. This has significant implications, especially in migratory contexts. For example, when chronically infected individuals move from hyperendemic areas like Huanta to low-endemic regions such as Lima (e.g., San Juan de Lurigancho), where most of the population is susceptible, there is a clear risk of HBV dissemination. To evaluate this risk, a cross-sectional study was conducted among residents of Lima’s “Huanta” settlement, which included both migrant and non-migrant families. Among 215 individuals assessed (130 females, 85 males), nine (4.2%) tested positive for HBsAg. Interestingly, five of these individuals were born in Lima, had never traveled to Huanta, were under 20 years old, and had no history of classical HBV risk factors (e.g., injectable drug use, blood transfusions, tattoos, surgeries, dental extractions, or sexual activity). These findings suggest that living with chronic HBV carriers from hyperendemic areas may facilitate horizontal transmission [42].

To further investigate this hypothesis, a seroepidemiological study was conducted among the household contacts of 126 HBsAg-positive index cases (children from Luricocha, Huanta). Among family members, 10.3% tested positive for HBsAg and 83.1% for anti-HBcAg. When stratified by kinship, siblings of the index cases showed the highest HBsAg positivity compared to other household members [32].

However, a subsequent nested case-control study, designed to assess the risk of intrafamilial HBV transmission in the Pampas River Valley using a previously described sample, yielded different results. Fifty relatives of schoolchildren with positive HBsAg serology (cases) were compared to 56 relatives of HBsAg-negative children (controls). Among relatives of cases, 13 (26.0%) were HBsAg-positive, compared to 18 (32.1%) among controls. Anti-HBcAg was detected in 70.0% of the case group and 83.9% of the control group. HBeAg was found in only three cases (6.0%) and one control (1.7%). No statistically significant differences in serological markers were observed between the two groups, leading to the conclusion that no increased risk of familial transmission could be demonstrated in this city [33].

In both hyperendemic and low-endemic regions, HBV carriers are suspected to play a key role in horizontal transmission, particularly among children [43,44]. However, this hypothesis still needs to be definitively proven and will require further investigation within the framework of molecular epidemiology.

## 8. Seroprevalence in High-Risk Groups

Hepatitis B virus (HBV) infection is known to disproportionately affect specific high-risk populations. Understanding the prevalence within these groups is essential for identifying targets for preventive interventions and tailored health screening programs.

In 1997, researchers collected 159 blood samples from military personnel stationed at “Fuerte Los Pokras” in Quicapata, Ayacucho. Four samples tested positive for HBsAg, indicating a prevalence of 2.52%. The positive individuals were between 16 and 25 years old; three were from Ayacucho and one from Lima. The most frequently reported risk factors included sexual promiscuity, tattoos, and overcrowding. Notably, cases of hepatitis C were also reported in this population [45].

In 2002, the prevalence of HBV infection was evaluated in 90 female sex workers attending the Regional Hospital of Ayacucho’s PROCET (Program for the Control of Sexually Transmitted Diseases and AIDS). The prevalence of HBsAg and anti-HBcAg was 11.1% and 26.7%, respectively. The age group most affected was 18–23 years (13.6% HBsAg and 26.7% anti-HBcAg). Higher prevalence was associated with a longer duration of sex work, a greater number of daily clients, and earlier onset of sexual activity. Most participants reported being originally from Lima (48.9%) and Huancayo (25.6%) [46].

Subsequently, between 2020 and 2021, a cross-sectional study was conducted at the Yanamilla Maximum Security Prison in Ayacucho to assess risk factors associated with HBV infection among inmates. A review of 270 medical records revealed an HBV infection frequency of 33.7% (91 cases). The main risk factors identified were sexual relations with high-risk individuals such as sex workers, drug use, and heterosexual orientation (68%) [47].

## 9. Vaccination and Prevention

Perinatal transmission constitutes the primary route of hepatitis B virus (HBV) infection in endemic populations [48]. Consequently, interrupting this transmission pathway is a critical first step toward eliminating HBV infection. In response to the significant burden of HBV in the Peruvian Andes, pilot studies for the inclusion of the HBV vaccine into the Expanded Program on Immunization (EPI) began in 1991 [29].

One of the earliest interventions took place in Abancay (2395 m above sea level) in the southern highlands of Peru. This region was classified as hyperendemic for both HBV and hepatitis D virus (HDV), where approximately 7% of all deaths were attributed to HBV-related complications [49].

This successful experience stimulated a similar intervention in 1994 in the province of Huanta, Ayacucho (2400 m above sea level), also considered hyperendemic for HBV and HDV [50]. In Huanta, about 8% of deaths were associated with severe liver diseases such as fulminant hepatitis, cirrhosis, or hepatocellular carcinoma. The vaccination campaign included 1412 children under 1 year of age and 5175 children aged 1 to 4 years. Each child received three doses of a recombinant HBV vaccine, following an EPI-adapted schedule: infants under 1 year received three doses over four months, while children aged 1–4 years received three doses over six months. After one year of implementation, 98.1% (n = 1386) of infants and 83.1% (n = 4353) of children aged 1–4 had completed the full three-dose series. Crucially, no serious adverse reactions were reported. Importantly, this campaign also significantly improved EPI coverage, which had declined in previous years—from 75% in 1991 to 64.5% in 1992 to 55.2% in 1993 [29].

In 1997, an evaluation of the HBV vaccination campaign in Huanta revealed a significant reduction in infection rates among children of the same age group, with prevalence dropping to 2.3–5.1% [22]. This confirmed that the inclusion of the HBV vaccine in the EPI was not only safe and effective in hyperendemic settings but also enhanced overall immunization coverage. Based on national and international evidence, in 1996, the Peruvian Ministry of Health formally incorporated the HBV vaccine into the EPI for all children under one year of age living in areas of moderate and high endemicity. At that time, approximately 120,430 children fell into this target group. This measure was driven by the increased risk of perinatal transmission in these regions and the availability of resources to support immunization. The policy aimed to expand vaccination to all children under five years of age and eventually to adolescents, particularly in hyperendemic areas such as the valleys of Huanta and the Apurímac River, where targeted adolescent vaccination began in 2002 [29,30].

Peru’s EPI now officially recommends hepatitis B vaccination (HepB) for children under 1 year of age, with the first dose administered at birth within 24 h (HepBB), followed by three additional doses (HepB3) at 2, 4, and 6 months of age as part of the pentavalent vaccine [24,26]. This policy, gradually introduced since 1991 and reinforced by the 2019 Ministry of Health Resolution (NTS Nº 159-MINSA/2019/DGIESP) for the prevention of mother-to-child transmission, forms a cornerstone of national HBV prevention efforts [27]. Birth-dose (HepBB) coverage has improved from 60% in 2003 to oscillating between 75% and 85% over the past decade [51,52], although challenges remain due to the logistical demands of timely administration, which have prevented further increases. By contrast, three-dose series (HepB3) coverage has consistently remained higher, ranging from 88 to 95% in 2010–2015, but has gradually declined to between 72% and 80% in recent years [51,52].

In Ayacucho, despite this historical progress, challenges persist. A retrospective study in 2019 assessed HBV immunization coverage among infants born to HBV-infected mothers at the Huanta Support Hospital between 2014 and 2018. Total HBV immunization coverage was 64.47%. While birth dose coverage reached 100%, administration of hepatitis B immunoglobulin (HBIg) was 77.78%, and the three-dose pentavalent vaccine coverage declined from 97.61% (1st dose) to 93.90% (2nd) and 80.26% (3rd dose) [53]. These figures fell below World Health Organization (WHO) recommendations for comprehensive HBV prophylaxis in neonates [25].

Further contributing to these challenges, knowledge gaps among caregivers regarding HBV transmission routes and consequences were identified in 72% and 84% of surveyed parents, and 53.3% of healthcare workers. Moreover, low adherence to EPI strategies (33.3%) and insufficient monitoring and evaluation (50%) were also reported as barriers to optimal vaccine delivery [54]. In response, educational interventions targeting secondary school students in Huanta have been implemented. Prior to the intervention, only 38.3% of students demonstrated knowledge of HBV, and 42.2% practiced preventive behaviors. Post-intervention evaluations revealed a dramatic increase to 95% in knowledge and 72.1% in preventive practices [55].

A temporal analysis of HBV incidence and vaccination coverage in Ayacucho between 2020 and 2024 reveals a clear inverse tendency (Figure 7). While vaccination coverage steadily declined from nearly 100% in 2020 to below 80% in 2024, HBV incidence showed a rising trend in several age groups. The most pronounced increase was observed among adults aged 30–59 years, whose incidence rose consistently throughout the period and peaked in 2023. Children (0–11 years) and young adults (18–29 years) also displayed upward trends, although with smaller absolute rates, suggesting ongoing community-level transmission despite the availability of immunization. In contrast, adolescents (12–17 years) maintained a relatively low and stable incidence, whereas the elderly (>60 years) experienced fluctuating rates with a moderate increase toward 2024.

Although HBV vaccination in Peru primarily targets children, the observed rise in incidence among adults likely reflects indirect, population-level effects. Declining vaccination coverage in younger cohorts increases the pool of susceptible individuals, sustaining viral circulation and consequently elevating the risk of exposure in historically unvaccinated adults. Studies have documented cohort effects, showing that the vaccination of adolescents can result in higher vaccination levels later observed among young adults [56], and that adolescent catch-up strategies reduce infection risk in the wider population [57]. Conversely, a French cohort demonstrated that the withdrawal of adolescent vaccination was associated with a rebound in HBV incidence, underscoring the importance of maintaining high coverage to preserve control achievements [58]. These findings highlight the urgent need to reinforce immunization programs in Peru, not only by sustaining high coverage in children, but also by expanding catch-up vaccination for adolescents and adults and strengthening outreach in rural and underserved areas where coverage remains most fragile.

In endemic areas where chronic HBV carriers are often asymptomatic, strengthening screening efforts is imperative. Active case-finding, comprehensive vaccination strategies, and targeted intervention programs for chronically infected individuals are essential pillars for HBV control and eventual elimination [59].

## 10. Conclusions

Ayacucho remains a high-burden HBV region in Peru, with historical peaks in mortality and infection. The introduction of HBV vaccination in the 1990s sharply reduced prevalence among children, demonstrating its safety and efficacy and supporting national EPI adoption. Despite vaccination, HBV persists in high-risk groups—military personnel, sex workers, inmates, and blood donors—revealing ongoing viral circulation. In recent years, reported HBV incidence has risen across different age groups, coinciding with declining vaccination coverage. This inverse relationship between vaccination and incidence suggests indirect population-level effects.

Challenges include knowledge gaps, low EPI adherence, and limited monitoring. Strengthening immunization, targeted screening, and interventions for high-risk groups are essential for controlling HBV in Ayacucho.

## Figures and Tables

**Figure 1 vaccines-13-00916-f001:**
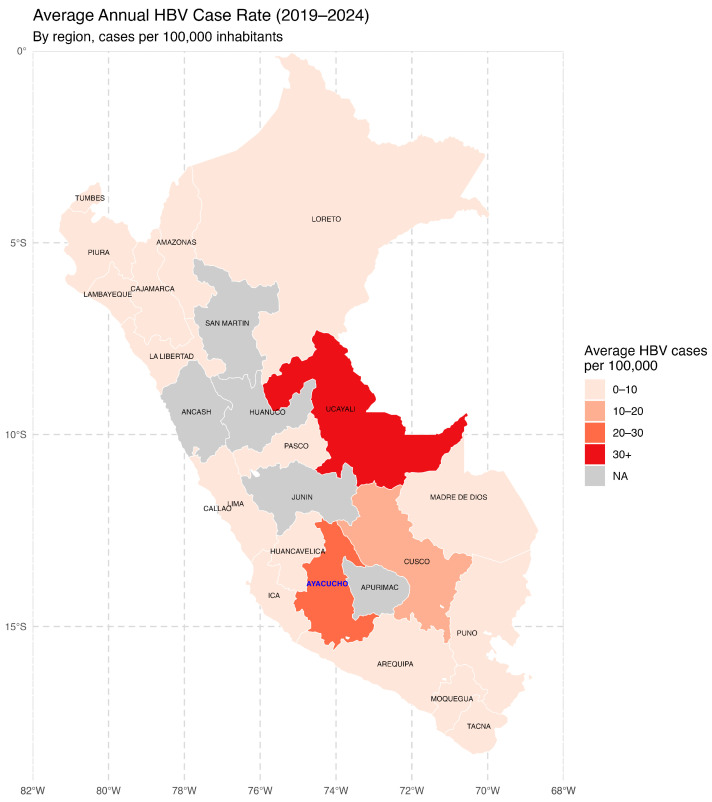
Average annual hepatitis B virus (HBV) incidence in Peru by region (2019–2024). The map illustrates the average annual rate of reported hepatitis B virus (HBV) cases per 100,000 inhabitants across Peruvian regions between 2019 and 2024. The regions are color-coded into four categories based on the incidence rate. The Ayacucho Region is shown in blue letters. NA: Not Available. Source: National Center for Epidemiology, Prevention and Disease Control—MINSA (Ministry of Health of Peru).

**Figure 2 vaccines-13-00916-f002:**
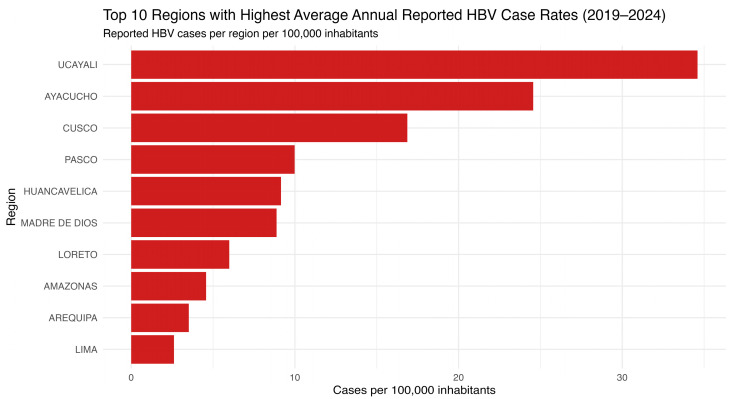
Top 10 Peruvian regions with the highest average annual HBV incidence (2019–2024). The horizontal bar chart presents the ten Peruvian regions with the highest average annual hepatitis B virus (HBV) case rates from 2019 to 2024, expressed per 100,000 population. Source: National Center for Epidemiology, Prevention and Disease Control—MINSA (Ministry of Health of Peru).

**Figure 3 vaccines-13-00916-f003:**
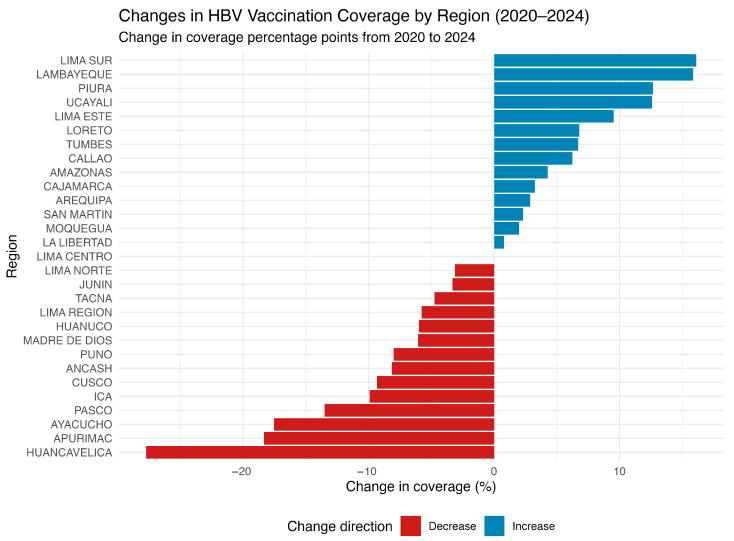
Change in HBV vaccination coverage by region in Peru (2020–2024). The horizontal bar chart illustrates the percentage change in hepatitis B virus (HBV) vaccination coverage across 29 Peruvian regions between 2020 and 2024. Each bar represents a region, ranked from greatest to smallest decline. Positive values (blue bars) indicate an increase in coverage, while negative values (red bars) indicate a reduction, relative to baseline levels in 2020. Source: National Center for Epidemiology, Prevention and Disease Control—MINSA (Ministry of Health of Peru).

**Figure 4 vaccines-13-00916-f004:**
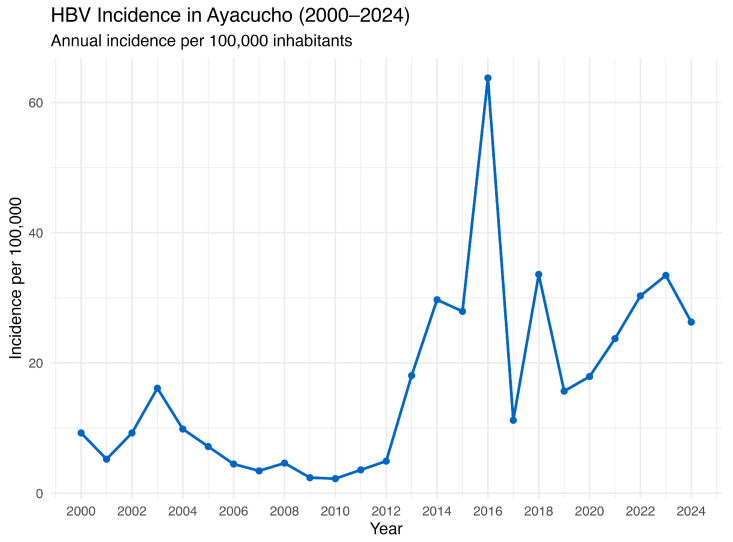
HBV annual incidence in Ayacucho, Peru (2000–2024). Source: National Center for Epidemiology, Prevention and Disease Control—MINSA (Ministry of Health of Peru).

**Figure 5 vaccines-13-00916-f005:**
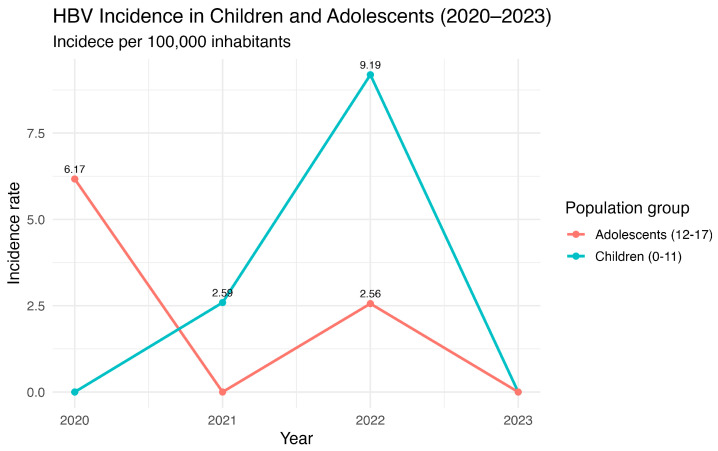
HBV annual incidence in children and adolescents in Ayacucho, Peru (2020–2023). Source: National Center for Epidemiology, Prevention and Disease Control—MINSA (Ministry of Health of Peru).

**Figure 6 vaccines-13-00916-f006:**
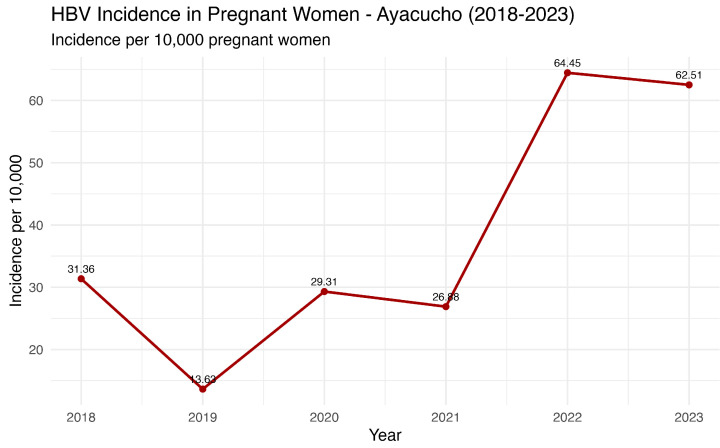
HBV annual incidence in pregnant women in Ayacucho, Peru (2018–2023). Source: National Center for Epidemiology, Prevention and Disease Control—MINSA (Ministry of Health of Peru).

**Figure 7 vaccines-13-00916-f007:**
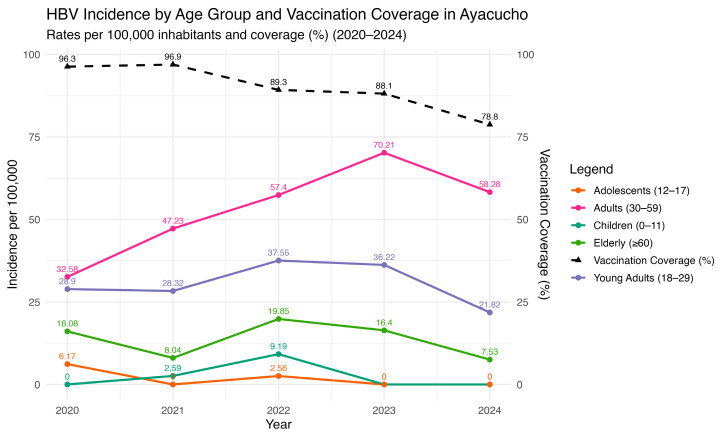
HBV incidence and vaccination coverage by age groups in Ayacucho, Peru (2000–2024). Source: National Center for Epidemiology, Prevention and Disease Control—MINSA (Ministry of Health of Peru).

## Data Availability

The information used to generate the figures was obtained from the National Center for Epidemiology, Prevention and Disease Control—MINSA (Ministry of Health of Peru) and is provided in the Appendix A.

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
