# Peer review of "Hepatitis B Virus Seroprevalence in Ayacucho, Peru: A Comprehensive Review Across the Pre-Vaccination and Post-Vaccination Periods"

_vaccines, 2025, doi:10.3390/vaccines13090916_

Round 1

Reviewer 1 Report

Comments and Suggestions for Authors

Ango-Aguilar, et al, reviewed existing epidemiological data on HBV seroprevalence in Ayacucho, Peru, emphasizing temporal changes observed before and after the implementation of vaccination programs to inform control strategies.

General concept comments

The critical finding stated in line 464-468 that declining immunization levels resulted in increased the incidence of HBV cases may not be sure in Ayacucho because the sustained increase in the number of HBV cases reported in Ayacucho starting in 2012, with notable peaks in 2016 and 2023, while HBV vaccination coverage marked decrease starting in 2021. Furthermore, the authors did not show the age of HBV cases. They should carry out a stratify analysis according to age. It is possible that these cases come from old population.

Specific comments 

1)No reference was found in the whole section of Epidemiological Overview of Hepatitis B Virus in Peru. Please provide all references.

2) It is unclear that the data in line 169 to 177 come from reference 18 or not. If so, please move the reference to the end of line 177. This applies to the whole paper.

3) Readers from Peru may be easy to understand the map in Fig 1. However, readers from other parts of the world may not understand where is it. Please add the name of each part to the map.

4)It is difficult to understand “HBV rate” in the title of Fig.2. HBV infection rate? HBV case? HBV chronic carrier? etc.

5) It will be better for readers to understand if authors add a figure to the section of 4 Seroprevalence in Childs according to timeline.  

6) The section of conclusion is too long.

7)The title looks like an article not review.

Author Response

We thank the reviewer for their thoughtful comments and constructive suggestions concerning our manuscript entitled “Hepatitis B Virus Seroprevalence in Ayacucho, Peru: A Comprehensive Analysis Across Pre- and Post-Vaccination Periods” (ID: vaccines-3778628), which enabled us to resubmit a clearly improved manuscript. We highlighted the amendments in the revised manuscript, and responded, point by point to, the comments listed below.

Reviewer #1:

Q0. Ango-Aguilar, et al, reviewed existing epidemiological data on HBV seroprevalence in Ayacucho, Peru, emphasizing temporal changes observed before and after the implementation of vaccination programs to inform control strategies.
R0. We greatly appreciate the comments and suggestions on our manuscript.

Q1. The critical finding stated in line 464-468 that declining immunization levels resulted in increased the incidence of HBV cases may not be sure in Ayacucho because the sustained increase in the number of HBV cases reported in Ayacucho starting in 2012, with notable peaks in 2016 and 2023, while HBV vaccination coverage marked decrease starting in 2021. Furthermore, the authors did not show the age of HBV cases. They should carry out a stratify analysis according to age. It is possible that these cases come from old population.
R1. We thank the reviewer for this observation. We have stratified the incidence data by age group and their interpretations in section 9. Since, as suggested, the incidence was not only in children but also in other age groups, we have modified the interpretations based on population-level effects. Based on all the adjustments, we have updated the conclusions. The changes are highlighted in yellow.

Q2. No reference was found in the whole section of Epidemiological Overview of Hepatitis B Virus in Peru. Please provide all references.
R2. We thank the reviewer for this observation. We've expanded the first paragraph to indicate the references and source of the data used in this section, as suggested. The changes are highlighted in yellow.

Q3. It is unclear that the data in line 169 to 177 come from reference 18 or not. If so, please move the reference to the end of line 177. This applies to the whole paper.
R3. We thank the reviewer for this observation. We have modified the reference at the end of the last sentence that references it. We've also done the same for all other references throughout the manuscript. The changes are highlighted in yellow.

Q4. Readers from Peru may be easy to understand the map in Fig 1. However, readers from other parts of the world may not understand where is it. Please add the name of each part to the map.
R4. We thank the reviewer for this observation. We have modified Figure 1 by labelling all the geographic regions on the map as suggested. We have also highlighted the Ayacucho region with blue letters. The new figure is now in the manuscript, and the changes to the legend are highlighted in yellow.

Q5. It is difficult to understand “HBV rate” in the title of Fig.2. HBV infection rate? HBV case? HBV chronic carrier? etc.
R5. We thank the reviewer for this observation. We have modified the title of the figure as “...HBV Case Rates” to make it more understandable as suggested.

Q6. It will be better for readers to understand if authors add a figure to the section of 4 Seroprevalence in Childs according to timeline.  
R6. We thank the reviewer for this observation. We have included a figure to show the incidence rates in children in recent years. In addition, we have added a paragraph describing the figure. The changes are now highlighted in yellow.

Q7. The section of conclusion is too long.
R7. We thank the reviewer for this observation. We have reduced the conclusions as suggested. The changes are highlighted in yellow.

Q8. The title looks like an article not review.
R8. We thank the reviewer for this observation. The title has been modified for review as suggested. The changes are now highlighted in yellow.

Reviewer 2 Report

Comments and Suggestions for Authors

The review presents the analysis of HBV seroprevalence and hepatitis B incidence in endemic territory against the background of universal vaccination. Presented data are very important as they show the importance of maintaining vaccination coverage to control virus circulation after the successful start of vaccination program. The manuscript is written well and may be of interest to a wide range of readers interested in the vaccination programs, as well as to healthcare policymakers. I have only some minor comments:

  1. Lines 32-33. Donors usually are not considered as population at-risk. Further in manuscript, data on HBV prevalence in donors are reasonably presented separately from data on risk groups.
  2. Section 2. Please clarify if incidence data presented here are combined data for cases of acute and chronic hepatitis B. Are cases of acute and chronic hepatitis B are registered separately or together in Peru?
  3. Figure 4. Please consider adding data on incidence presented as cases per 100,000 population together with total number of cases per year.
  4. Lines 202-209. Please discuss here coverage with HBV laboratory diagnostics and HBsAg tests used in the country throughout the years. Who are routinely tested for HBsAg in the country? Could expansion of HBsAg screening or switch to more sensitive HBsAg tests impact the number of registered cases after 2012?
  5. Line 214. Seroprevalence in Children
  6. Line 393. Please remove “DNA” from vaccine description.
  7. Section 9. Please clarify whether Expanded Program on Immunization suggest HBV vaccination of children under one year with first dose given at birth or not. If birth-dose is recommended, please provide data on birth-dose coverage, if such data are available.

Author Response

We thank the reviewer for their thoughtful comments and constructive suggestions concerning our manuscript entitled “Hepatitis B Virus Seroprevalence in Ayacucho, Peru: A Comprehensive Analysis Across Pre- and Post-Vaccination Periods” (ID: vaccines-3778628), which enabled us to resubmit a clearly improved manuscript. We highlighted the amendments in the revised manuscript, and responded, point by point to, the comments listed below.

Reviewer #2:

Q0. The review presents the analysis of HBV seroprevalence and hepatitis B incidence in endemic territory against the background of universal vaccination. Presented data are very important as they show the importance of maintaining vaccination coverage to control virus circulation after the successful start of vaccination program. The manuscript is written well and may be of interest to a wide range of readers interested in the vaccination programs, as well as to healthcare policymakers.
R0. We greatly appreciate the comments and suggestions on our manuscript.

Q1. Lines 32-33. Donors usually are not considered as population at-risk. Further in manuscript, data on HBV prevalence in donors are reasonably presented separately from data on risk groups.
R1. We thank the reviewer for this observation. We have modified the sentences in lines 32-33 to differentiate high-risk populations from blood donors, as noted. The changes are now highlighted in yellow.

Q2. Section 2. Please clarify if incidence data presented here are combined data for cases of acute and chronic hepatitis B. Are cases of acute and chronic hepatitis B are registered separately or together in Peru?.
R2. We thank the reviewer for this observation. In Peru, although surveillance is based on individual case reporting, there are no explicit separate categories in official reports that distinguish between acute and chronic infection. The definition of the confirmed cases analyzed, as well as the clarification that both acute and chronic infections are included, has been inserted in the first paragraph of section 2. The changes are highlighted in yellow.

Q3. Figure 4. Please consider adding data on incidence presented as cases per 100,000 population together with total number of cases per year.
R3. We thank the reviewer for this observation. We have modified Figure 4 by adding a graph showing the incidence of HBV per 100,000 population, as suggested. To maintain uniformity among all figures, we have converted the others into terms of incidence. Changes to the figure legend are highlighted in yellow.

Q4. Lines 202-209. Please discuss here coverage with HBV laboratory diagnostics and HBsAg tests used in the country throughout the years. Who are routinely tested for HBsAg in the country? Could expansion of HBsAg screening or switch to more sensitive HBsAg tests impact the number of registered cases after 2012?.
R4. We thank the reviewer for this observation. We have added some lines to discuss, with references, that an increase in HBsAg screening and the implementation of more sensitive methods may have contributed to the increase in case reporting. The changes are now highlighted in yellow.

Q5. Line 214. Seroprevalence in Children.
R5. We thank the reviewer for this observation. We have changed the word Childs to Children as suggested. The changes are now highlighted in yellow.

Q6. Line 393. Please remove “DNA” from vaccine description. 
R6. We thank the reviewer for this observation. We have removed the term DNA as suggested. The changes are now highlighted in yellow.

Q7. Section 9. Please clarify whether Expanded Program on Immunization suggest HBV vaccination of children under one year with first dose given at birth or not. If birth-dose is recommended, please provide data on birth-dose coverage, if such data are available.
R7. We have included a paragraph detailing the HBV vaccination program in Peru, as well as references to the coverage achieved by each type of vaccination nationwide. The changes are highlighted in yellow.

Reviewer 3 Report

Comments and Suggestions for Authors

The authors of this manuscript have characterized infections by hepatitis B virus (HBV) in a endemic region in Peru before and after the introduction of vaccination.  The results are interesting and worth publication.  In particular the relationship between the level of vaccination and the incidence of infection clearly shows the effectiveness of the vaccine.  I have only two minor points for correction.  Figure 1 would be improved by labeling of the areas of high incidence.   It would be useful to know exactly where Ayacucho is located.  Second, Figure 3 is titled Decline in HBV etc.  Since there are also increases shown in this figure it would be more accurate to title the figure Changes in HBV etc.  

Author Response

We thank the reviewer for their thoughtful comments and constructive suggestions concerning our manuscript entitled “Hepatitis B Virus Seroprevalence in Ayacucho, Peru: A Comprehensive Analysis Across Pre- and Post-Vaccination Periods” (ID: vaccines-3778628), which enabled us to resubmit a clearly improved manuscript. We highlighted the amendments in the revised manuscript, and responded, point by point to, the comments listed below.

Reviewer #3:

Q0. The authors of this manuscript have characterized infections by hepatitis B virus (HBV) in a endemic region in Peru before and after the introduction of vaccination.  The results are interesting and worth publication. In particular the relationship between the level of vaccination and the incidence of infection clearly shows the effectiveness of the vaccine. I have only two minor points for correction.

R0. We greatly appreciate the comments and suggestions on our manuscript.

Q1. Figure 1 would be improved by labeling of the areas of high incidence. It would be useful to know exactly where Ayacucho is located.
R1. We thank the reviewer for this observation. We have modified Figure 1 by labelling all the geographic regions on the map as suggested. We have also highlighted the Ayacucho region with blue letters. The new figure is now in the manuscript, and the changes to the legend are highlighted in yellow.

Q2. Second, Figure 3 is titled Decline in HBV etc.  Since there are also increases shown in this figure it would be more accurate to title the figure Changes in HBV etc.
R2. We thank the reviewer for this observation. We have modified the title of Figure 3 as suggested. We have also changed the bar colors to differentiate between increases and decreases. The changes in the legend are highlighted in yellow.

Round 2

Reviewer 1 Report

Comments and Suggestions for Authors

No further comments.